# Semantic Priming and Its Link to Verbal Comprehension and Working Memory in Children with Learning Disorders

**DOI:** 10.3390/brainsci13071022

**Published:** 2023-07-01

**Authors:** Benito Javier Martínez-Briones, Thalía Fernández, Juan Silva-Pereyra

**Affiliations:** 1Facultad de Estudios Superiores Iztacala, Universidad Nacional Autónoma de México, Tlanepantla 54090, Mexico; benjavik332@gmail.com; 2Departamento de Neurobiología Conductual y Cognitiva, Instituto de Neurobiología, Universidad Nacional Autónoma de México Campus Juriquilla, Querétaro 76230, Mexico; thaliafh@yahoo.com.mx

**Keywords:** learning disorders, event-related potentials, semantic priming, working memory, children

## Abstract

Children with learning disorders (LD children) often have heterogeneous cognitive impairments that affect their ability to learn and use basic academic skills. A proposed cause for this variability has been working memory (WM) capacity. Altered patterns of event-related potentials (ERPs) in these children have also been found in the N400 component associated with semantic priming. However, regarding the semantic priming effect in LD children, no distinction has been made for children with varying WM abilities. This study aims to explore the relationship of WM with the brain’s electrophysiological response that underlies semantic priming in LD children that performed a lexical decision task. A total of 40 children (8-10 years old) participated: 28 children with LD and 12 age-matched controls. The ERPs were recorded for each group and analyzed with permutation-based t-tests. The N400 effect was observed only in the control group, and both groups showed a late positive complex (LPC). Permutation-based regression analyses were performed for the results from the LD group using the WISC-IV indices (e.g., Verbal Comprehension and WM) as independent predictors of the ERPs. The Verbal Comprehension Index, but not the WM index, was a significant predictor of the N400 and LPC effects in LD children.

## 1. Introduction

In children, learning disorders (LDs) are diagnosed after persistent difficulties (at least 6 months) in learning the basic academic skills of reading, writing, and/or mathematics. Children with LDs have standardized scores on academic tests that are significantly below the expected for their age, and their learning difficulties substantially interfere with school performance or activities of daily living. These difficulties cannot be explained by intellectual disabilities, inadequate school instruction, or sensory/neurological impairments [1].

The study of LDs focuses mostly on specific disorders of reading (such as dyslexia) or subtypes of combined learning disorders, formerly known as “learning disorders not otherwise specified” [2]. The prevalence of LDs is 5–20% in children and adolescents, making it one of the most common neurodevelopmental disorders [1,3,4,5]; if a specific reading disorder is the most prevalent subtype, it can appear in combination with other disorders (writing or mathematics) in up to 80% of LD cases. Thus, in practice, LDs are fairly heterogeneous in the academic domains they affect [1,6].

A combination of learning disorders is accompanied by a heterogeneous frame of cognitive impairments in these children, with different processes recognized as affected, such as verbal comprehension, processing speed, and working memory (WM) [6,7,8]. Working memory is the most consistently affected process in these children [9,10] and can predict future academic difficulties [11]. A defective WM means a diminished capacity to access, maintain, or retrieve information. Children require an adequate working memory capacity to properly develop their basic academic skills [12]; thus, an impaired WM has been proposed as a common factor underlying the defective acquisition of academic skills in individuals with LDs [12,13].

The electroencephalogram (EEG) has been a powerful tool for identifying neural correlates of LDs. The power spectrum of the resting-state EEG of these children shows abnormally slower EEG activity than that of age-matched control children with typical development, with more theta power in frontal regions and less alpha power in posterior sites, indicating an apparent lag in the brain’s functional development that is similar to what would be expected in a younger healthy child [14,15,16].

Task-related EEG power-spectrum analyses have been carried out on the performance of Sternberg WM tasks [17,18]. Children with LDs, compared to controls, showed overall slower task-related activity with higher delta and theta power and lower gamma power at posterior sites [19]. Such patterns of activity indicate inefficient neural recruitment that requires more effort from the child to achieve proper cognitive performance because higher delta power relates to sustained concentration coupled with the inhibition of sensory information [20,21,22], higher theta power indicates the recruitment of more neural resources in less-apt individuals or during more difficult conditions [16,23,24], and lower gamma power reflects a defective binding of memory representations [25,26,27].

A more robust approach to analyzing task-related EEGs uses the technique of event-related potentials (ERPs). ERPs enable researchers to study the time-domain aspects of cognitive processing by averaging electrophysiological patterns that are time-locked to external or internal stimuli [28]. Different ERP components with specific latency, amplitude, and topography have been recognized in specific cognitive tasks. In particular, children with LDs have been found to have altered or absent ERP components during the performance of several tasks that emphasize different cognitive abilities, such as working memory [29], arithmetic calculation [30,31], and semantic processing [32,33].

In WM tasks, children with LDs compared to controls were found to have a greater and delayed P300 effect (that normally peaks at around 300 ms after stimulus onset). This finding is understood as indicative of these children requiring more neural resources for attention allocation and the updating of memory representations [29,34].

In a study of children with specific disorders of mathematics performing an arithmetic verification task, an absence of both an N400 component (negative peak at around 400 ms after stimulus onset) and a late positive complex (LPC: positive-going waveform between 500 and 900 ms after stimulus onset) was found in comparison to control children [30]. In this task, the N400 component would be elicited with a greater amplitude for incongruent results of an arithmetic operation compared to congruent results; this difference in amplitudes is considered an arithmetic N400 effect [35]. Moreover, an LPC effect may be indicative of a re-evaluation process of the arithmetic operation [36,37]. Both of these effects, in general, appear more pronounced in individuals with better arithmetic skills [38]. Moreover, in the study mentioned above by Cárdenas et al. [30], a positive correlation was found between WM ability and the LPC effect in the LD group; thus, WM was reiterated as a significant contributor to the heterogeneity of LDs and cognitive deficits in these children.

However, the above-mentioned N400 effect of incongruity versus congruity is of a more general nature of sense-making and has been studied mainly in the context of verbal abilities with the phenomenon of semantic priming as an expectancy; the N400 effect and the LPC are used as indices of such priming. The former indicates semantic predictability as the ease with which a word can be integrated into the context and is typically maximal over the left centro-parietal scalp regions, while the LPC is maximal over the frontal regions and reflects an expectancy violation, which is when a stimulus does not match our expectations. The LPC may also be involved in adjusting our internal predictive capacities to foresee what will happen next [39,40].

As a general characteristic, lexical decision tasks include words and pseudowords (legible but meaningless strings of letters), and the individual must distinguish between them. In particular, lexical decision tasks designed to study the phenomenon of semantic priming should include two conditions: (1) words (primed) whose processing is facilitated by the prior presentation of a semantically related word, and (2) words (non-primed) whose processing is not facilitated by the prior word; the N400 effect appears when the non-primed condition is significantly more negative. In studies of children with LDs that performed semantic priming tasks, this N400 effect (of around 350–600 ms after stimulus onset) was absent compared to age-matched controls [32,33,41]. However, the LPC (with a peak of 600 ms after stimulus onset), thought to be evoked when subjects are unable to integrate a word into their prior situation and resulting in a reallocation of attention and reprocessing of the input [40], has indeed been found in both children with LDs and age-matched controls [33]; thus the N400 effect is considered significantly impaired in LDs, and such impairment may indicate possible problems with verbal comprehension [33,42,43].

Compared to the more precise topography and laterality of the N400 and LPC components in healthy adults [40,43], children with LDs have a broader and relatively diffuse topography of the components [32,33], including an absent hemispheric asymmetry in children with LD compared to controls [44], with a notable exception being an impaired N400 effect in lateral parietal and occipital sites [41]. Such diffuse topography of the ERP components may be related to the overall heterogeneity of LDs and cognitive impairments, in that a child with LD requires more pronounced neural recruitment to understand the meaning of words and concepts [30,32,44].

Because there seem to be common cognitive deficits underlying LDs [13,45], with WM highlighted as a main culprit next to processes such as verbal comprehension and processing speed, the objective of the present work is, first, to replicate an impaired N400 effect in children with LDs while they perform a lexical decision task and, second, to explore the relationship of Verbal Comprehension and WM (indices of the Wechsler Intelligence Scale for Children 4th Edition, WISC-IV [46]) as independent predictors of the N400 and LPC components, which serve as indices of the semantic priming effect in children with LDs.

Verbal comprehension might be specifically related to the N400 component because the latter is an index of semantic processing [33,43,47]. Working memory capacity may especially affect the extent to which semantic priming occurs, which relies on WM to temporarily store the prior stimulus and use it to activate the related information in long-term memory. Thus, individuals with a higher WM, being better able to maintain and manipulate primed stimuli, could have stronger N400 effects [30,44,48]. Additionally, the LPC has previously been found to be related to memory retrieval in healthy adults [47,49]. Accordingly, we expected to find a direct relationship between a higher WM ability and the N400 and LPC effects in children with LDs.

## 2. Materials and Methods

### 2.1. Participants

This study’s protocol complies with the Ethical Principles for Medical Research Involving Human Subjects established by the Declaration of Helsinki [50]. All participants and their parents signed informed consent forms that were approved by the Ethical Committee of the Facultad de Estudios Superiores Iztacala (CE/FESI/102022/1558) of the Universidad Nacional Autónoma de México (UNAM).

Forty right-handed children aged 8-11 years were selected from elementary schools. All children met the following inclusion criteria: (1) a normal neurological and psychiatric assessment (except for the LD diagnostic requirements, stated below) without language impairments or visual/hearing acuity problems (those with visual problems used correcting glasses); (2) an intelligence quotient (IQ) above 70 (Wechsler Intelligence Scale for Children 4th Edition, WISC-IV [46]), to exclude children with intellectual disability); and (3) without severe socioeconomic disadvantages, that is, a mother (or tutor in her absence) that has at least completed elementary school education and has a per capita income greater than 50 percent of the minimum wage.

Twenty-eight children (thirteen girls) were diagnosed with an LD, according to the following three criteria: (a) poor academic achievement reported by teachers and parents; (b) percentiles of 10 or lower in the subscales of reading, writing, or mathematics of the Infant Neuropsychological Scale for Children [51]; and (c) the final decision of LD was delivered by a psychologist according to the DSM-5 criteria for LDs [52]. Some children failed to complete a few items of an attentional evaluation but did not meet the DSM-5 criteria for ADHD [52].

The frequencies of learning disorders found in our LD sample were as follows: seven children were impaired in all three domains (reading, writing, and mathematics); four children were impaired in reading and writing; five children were impaired in reading and mathematics; two children were impaired in writing and mathematics; six children were impaired in reading; one child was impaired in writing; and three children were impaired in mathematics (Figure 1). We acknowledge a lack of homogeneity in our sample of children with LDs; however, WM deficits have been found for all LD subtypes [53,54], which is in line with our aims.

Twelve children (seven girls) formed a control group that had good academic achievement according to their parents and teachers; they also obtained percentiles of at least 37 in the reading, writing, and mathematical domains of the Infant Neuropsychological Scale for Children [51]. Table 1 shows the main descriptive characteristics of both groups.

### 2.2. Indices of the WISC-IV Scale

The WISC-IV consists of 15 subtests grouped into 4 indices: Verbal Comprehension Index, Perceptual Reasoning Index, Working Memory Index, and Processing Speed Index [46]:Verbal Comprehension Index: Measures a child’s verbal abilities, such as vocabulary, comprehension, and general knowledge. The subtests that support this index are Similarities, Vocabulary, and Comprehension.Perceptual Reasoning Index: Measures a child’s nonverbal reasoning abilities, such as visual–spatial skills, perceptual organization, and problem solving. The subtests that support this index are Block Design, Picture Concepts, and Matrix Reasoning.Working Memory Index: Measures a child’s working memory ability, i.e., the capacity to hold and manipulate information for short periods of time. The subtests that support this index are Digit Span and Letter–Number Sequencing.Processing Speed Index: Measures a child’s processing speed abilities, such as the speed of performing simple tasks. The subtests that support this index are Coding and Symbol Search.

In children with LDs, there are instances of deficits in verbal comprehension, working memory, and processing speed [6,7,8]; however, in this study, the four indices (including perceptual reasoning) were taken as independent regressors of the ERP data described below, although according to the literature [33,43,44,47], we only expected to find WM and verbal comprehension as significant predictors of the ERP components.

### 2.3. Lexical Decision Task and ERP Acquisition

In a lexical decision task [55], participants are presented with a continuous sequence of letter strings, which can be either real words (e.g., “cow”, “duck”, “grape”) or nonsense words (e.g., “pilk”, “sug”, “verp”). Children are instructed to respond as quickly and accurately as possible as to whether each string of letters is a real word or not by pressing a button if they believe the letter string is a real word or pressing another button if they believe it is not (in this study, the button for each type of response was pressed with different thumbs and counterbalanced across subjects).

The real words belonged to two main categories: animals or vegetables. An initial alternate version of the task was piloted with a sample of 10 children with typical development that had to decide if 100 words about animals and the same number about vegetables were real words. Fifty correctly recognized words from each semantic category were taken as the real words for the current version of the task, and twenty-five words from each category whose letters were permutated to obtain nonsensical but readable pseudowords were used as the nonsense words.

Because semantic priming is a phenomenon in which the processing of a target word is facilitated by the prior presentation of a related word (i.e., of the same semantic category), resulting in faster response times and increased accuracy compared to non-related words, we thus isolated two main conditions: primed stimulus if the second word of a sequence followed a semantically related word (e.g., ‘duck’ after ‘cow’); and non-primed stimulus if the second word was a semantically unrelated word (e.g., ‘grape’ after ‘cow’). The conditions were thus compared by subtracting the primed from the non-primed stimuli to obtain the semantic priming effect. The pseudowords were not considered for the analysis, as they were only used to verify that the children correctly identified the words. Each stimulus lasted 2200 ms and was presented sequentially and pseudorandomly across conditions with an interstimulus interval of 500 ms and ~50 trials per condition. Figure 2 shows the experimental flow chart that includes an illustration of the task.

To obtain the behavioral data, the median reaction times (RTs) of correct responses for each condition were calculated for each subject. The percentage of correct responses for each condition was transformed using the inverse of the sine function (arcsine) [square root (percentage/100)] to ensure a normally distributed dataset [56].

A 19-channel EEG (Ag/AgCl electrodes held in position with a cap according to the 10–20 International System; Electro-Cap International Inc., PO, OH, USA), referenced to linked earlobes (A1A2), was recorded with a MEDICID IV system (Neuronic S.A.; Mexico City, Mexico) and a Track Walker v5.0 data system while the child was performing the task. The bandwidth of the amplifiers was 0.5–50 Hz, with a sampling rate of 200 Hz. Blinking and eye movements were monitored from a supra-orbital electrode and from an electrode placed at the external canthus of the right eye.

The raw EEG data were preprocessed with the EEGLAB software [57]. With an independent component analysis (ICA)-based decomposition [58], the components that seemed to be artifacts due to eye movements or excessive muscle activity were rejected. Then, artifact-free EEG segments with a duration of 800 ms and a pre-stimulus baseline of 200 ms and synchronized with the primed and non-primed words were selected. Approximately equal numbers of EEG segments with correct answers were acquired for each condition (primed mean segments = 33.5 ± 5.9; non-primed mean segments = 33.2 ± 5.9) and included in the ERP averages across subjects.

### 2.4. Statistical Analysis

ERPs were obtained for each group (control and LD) and each experimental condition. The topography of the priming-related components in children with LDs has previously shown notable variability compared to that in age-matched controls [33,44], not to mention compared to that reported in the priming literature on healthy adults. Thus, we did not select any ROIs in order to capture the variance across all channels simultaneously. Figure 3 and Figure 4 show the grand average ERPs of the groups. It can be noted by visual inspection that in the control group, from approximately 400 to 800 ms over most of the channels, the brainwaves associated with non-primed words were negatively larger than those associated with primed words, thus behaving as typical N400 and LPC components. These brainwaves first showed a negative peak at approximately 400 ms (N400) that turned into a positive deflection (LPC) after 500 ms, which is fairly similar to the patterns reported by others [32,33]. Thus, the N400 component was considered the mean amplitude within the interval of 355–550 ms, and the LPC was defined as the mean amplitude within 555–800 ms.

Nonparametric permutation-based t-tests (5000 permutations) were applied for the comparisons between groups in terms of age, academic performance (reading, writing, and mathematics), the WISC-IV indices, and the ERP components using a statistical tool from the eLORETA software [59].

Within-condition comparisons (primed vs. non-primed) were performed for the behavioral data (response times and correct responses) and over the 19 channels of the ERP data for each of the two N400 and LPC components. To establish possible relationships between the WISC-IV cognitive indices and the ERPs in the LD group, nonparametric permutation-based regression analyses were performed between each separate WISC-IV index as regressors of the N400 and LPC components. These nonparametric techniques do not require a theoretical distribution because the null-hypothesis distribution of statistical tests is iteratively generated by shuffling the data, and no corrections for multiple comparisons are needed when several channels or time points are assessed [60]. (See also Figure 2 for additional information).

## 3. Results

According to the main descriptive results (Table 1), there were no significant differences between groups in the ages of the participants. However, the control group, as expected, showed significantly better scores for the Full-Scale IQ and the academic performance variables of reading, writing, and mathematics compared to the LD group.

### 3.1. Cognitive Performance and Behavioral Results of the Lexical Decision Task

Table 2 shows that the children with LDs, compared to the controls, were impaired in the WISC-IV indices of Verbal Comprehension and WM. No significant differences were found in the indices of Perceptual Reasoning or Processing Speed.

Table 3 summarizes behavioral performance, showing the percentage of correct responses and response times for both groups. The control group showed significantly more correct responses (accuracy) and faster response times in the primed condition compared to the non-primed condition. The LD group only showed faster response times, but no differences were found in their within-group accuracy comparison.

### 3.2. ERP Results: Within-Condition and Between-Group Comparisons

Regarding the within-condition comparison for each group, the control group showed an overall strong semantic priming effect with significant N400 (range of t values = [−2.263, −2.256], *p* < 0.027) and LPC effects (range of t values = [−2.854, −3.207], *p* < 0.012) for all 19 channels, which are the expected patterns of activity [32,33,44]. The LD group showed only a significant LPC effect (range of t values = [−2.227, −2.748], *p* < 0.022) for all 19 channels, which is indicative of an impaired semantic priming effect, a finding also in line with previous work.

The between-group comparison did not reveal significant differences in the ERP components. This finding might be due to considerable variability in the brain responses of children with subtypes of LD [44,61]; thus, some possible significant differences may have been annulled. Additionally, the control group consisted of only 12 children, and this particular between-group comparison might have been underpowered. The within-condition results point to a picture in which the control group shows an expected semantic priming effect that is deficient in the LD group. This finding, coupled with the fact that the LD group did show impaired performance in the indices of Verbal Comprehension and Working Memory, does not affect the main aim of the regression analysis for the LD group.

### 3.3. Relationship between Cognitive Performance and the N400 Effect in the LD Group

The four indices of the WISC-IV scale were thus taken as independent regressors for the N400 and LPC effects in the LD group. We found that only higher Verbal Comprehension was a significant direct predictor of both the N400 and LPC effects in the LD group for all 19 channels, as shown in Table 4. No significant relationship was found between the other three indices (WM, Perceptual Reasoning, and Processing Speed) and the N400 or LPC effects. WM, as one of the impaired cognitive processes found in children with LDs, was expected to be a significant predictor of an impaired priming effect. Such non-significant findings do not support the hypothesis that the symptomatology of LDs is highly dependent on WM deficits.

Figure 5 shows a regression map using only the Verbal Comprehension Index as a predictor of the N400 and LPC effects for the LD group. Although the regression results were significant for all channels, it can be appreciated that Verbal Comprehension seems to predict a higher N400 effect more significantly in the right hemisphere and a higher LPC effect in the left hemisphere. Because children with LDs do seem to have less hemispheric asymmetry and an impaired N400 component with an intact LPC [33,44], the regression involving the N400 effect might be related to a compensatory process of contralateral recruitment of neural resources [62].

## 4. Discussion

In the present work, we aimed to (1) replicate the finding of an altered N400 semantic priming effect in children with LDs and (2) explore the relationship between indices of cognitive performance and the N400 and LPC effects, emphasizing the possible relationship between priming and WM, the latter being a proposed main culprit for the learning disorders and cognitive deficits in these children.

Children with LDs have been previously found to have impairments in cognitive domains such as verbal comprehension, working memory, and processing speed [6,7,8]. The WISC-IV scale gives an index for each of these domains and an index for perceptual reasoning. In this study, children with LDs were impaired in Verbal Comprehension and WM indices compared to age-matched controls. In the LD group, the indices were then used as predictors of performance in the lexical decision task.

Regarding behavioral performance, the control children showed significantly more correct responses and faster response times in the primed condition compared to the non-primed condition; however, the LD group showed only faster response times in the primed over the non-primed condition. Thus, poorer performance in the lexical decision task may already be present at this behavioral level, as can be seen by a certain lack of discernment of conditions, with similar performance errors in the primed and non-primed conditions. Nonetheless, the faster response times for the primed condition show at least a weaker behavioral semantic priming effect in children with LDs compared to children with typical development.

The N400 and LPC components, associated with semantic priming, were thus explored. The control group showed significant N400 and LPC effects in their within-condition comparison for all 19 channels, hence displaying an overall strong semantic priming effect. However, the LD group showed only a significant LPC effect (for all 19 channels). This finding has successfully replicated the result of previous work pointing to an atypical N400 component that, because this component is an index of semantic prediction, could reflect problems with verbal comprehension in these children [33,42]. On the other hand, the intact LPC, indicating expectancy violations, was also expected and might reveal effortful attempts at comprehension by children with LDs.

Building on the above-mentioned results, and by taking the four cognitive indices of the WISC-IV scale as independent regressors for the N400 and LPC components in the LD group, we expected to find a significant direct relationship of the Verbal Comprehension and WM indices with higher N400 and LPC effects. No significant relationship was found between WM and these indices of semantic priming; however, a higher Verbal Comprehension score was indeed a significant direct predictor of both the N400 and LPC effects for all channels. This suggests that a child with better verbal comprehension exhibits a stronger predictive semantic capacity that manifests as easier integration of words based on the context, coupled with the concurrent adjustment of such predictions after expectancy violations.

Because symptoms of LD have been proposed to be directly due to a WM deficit [13,63], semantic priming is considered to rely on WM in that a primed stimulus is temporarily maintained and activates related information in long-term memory [47,49], and previous work has highlighted a relationship between WM capacity and the ERPs associated with an arithmetic verification task [30]; such sets of findings point to a picture of WM having great explanatory power in characterizing a learning disorder, which includes defective semantic priming as an example of a cognitive deficit in children with LDs.

However, our results alternatively suggest that WM does not hold a crucial role in causally accounting for the semantic priming effect. In line with previous work [33,42], the weak semantic priming effect may be better explained by poor verbal comprehension, another main domain of cognitive performance that might have an independent influence in predicting part of the heterogeneity of learning disorders [6,63]. Thus, poor verbal comprehension could be related to a defective N400 component that in turn reflects a subpar predictive semantic capacity [33,43,47]. Moreover, although WM has been proposed as being involved in the priming phenomenon, the only direct finding of a relationship between a better WM and a greater priming effect in children with LDs was reported in a study on arithmetic priming [30]. The discrepancy between our results and those of that study may be due to a stronger relationship between WM and arithmetic than with semantic skills. Also, a significant difference between the task employed by Cárdenas et al. [30] and ours should be noted. In that case, the two items that determined the priming effect were of a different nature, i.e., an arithmetic operation and a written number, while in this work, the two items that determined the priming effect were of the exact same nature (both were words). Thus, the influence of WM in learning disorders may be fairly distant from a pure semantic priming effect, as WM seems to mainly affect the ability of individuals to carry out more complex processes such as reading and arithmetic operations. In line with this, a study [63] investigating the association between WM and reading ability found that other affected factors in LDs, such as fluid intelligence, verbal abilities, and phonological awareness, did not mediate such association. Instead, WM seems to predict a learning disorder of reading or mathematics, independently of verbal, i.e., phonological and semantic, skills.

More research is needed to elucidate the role of cognitive deficits such as verbal comprehension, working memory, and processing speed in learning disorders. To further clarify the relative role of such cognitive domains as some of the main contributors to the symptomatology of LDs, we propose that a more thorough relationship should be traced between indices of cognitive performance and the activity of different networks of the EEG resting state because several studies have proposed that resting-state measures are useful predictors of cognitive performance in different kinds of samples, from children [64] to healthy adults [65] and elderly individuals [66,67].

## 5. Conclusions

The N400 and LPC components in children with LDs responding to a lexical decision task were investigated as indices of semantic priming. The N400 effect, reflecting semantic processing, was impaired in these children, while the LPC, associated with memory retrieval and expectancy violation, appeared to be intact. In exploring the relationship between these children’s affected cognitive domains and such ERP components, verbal comprehension, but not working memory, was found to be an important predictor of the N400 and LPC effects in these children. These results indicate that children with LDs with better verbal comprehension exhibit a stronger predictive semantic capacity, coupled with a more effective adjustment of their predictions after expectancy violations.

## Figures and Tables

**Figure 1 brainsci-13-01022-f001:**
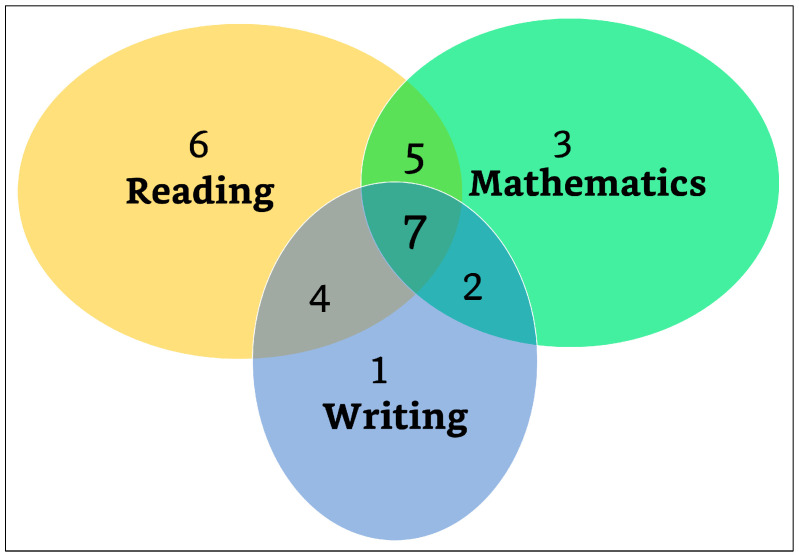
Venn diagram of the distribution of learning disorders.

**Figure 2 brainsci-13-01022-f002:**
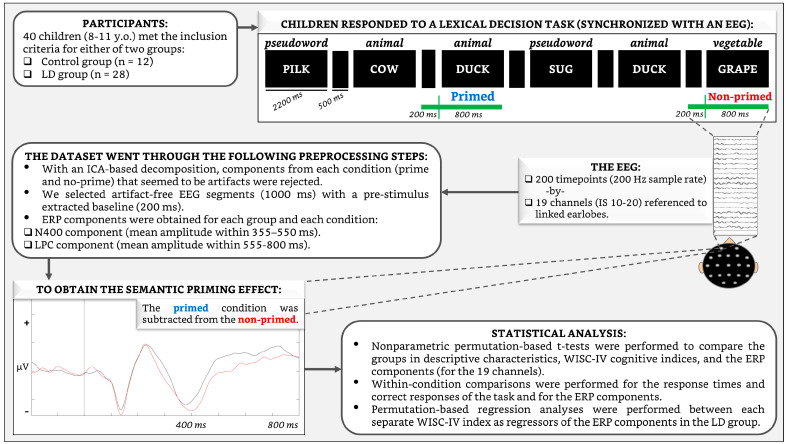
Experimental flow chart with an illustration of the lexical decision task.

**Figure 3 brainsci-13-01022-f003:**
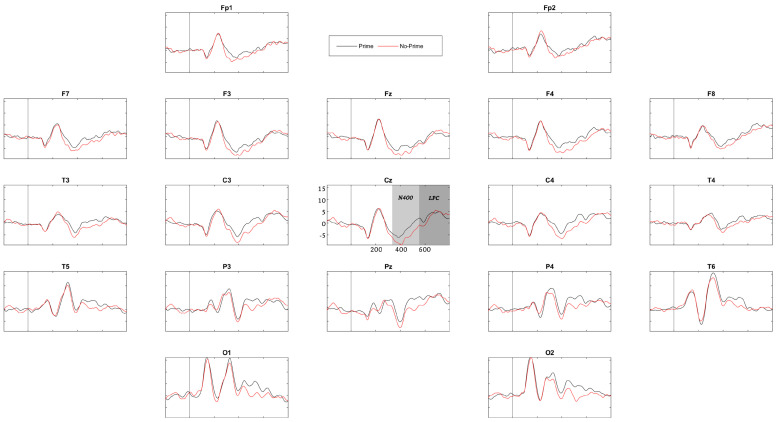
ERPs of the control group. The N400 time range (355–550 ms) is represented in the Cz channel in pale gray. The LPC time range (555–800 ms) is represented in dark gray.

**Figure 4 brainsci-13-01022-f004:**
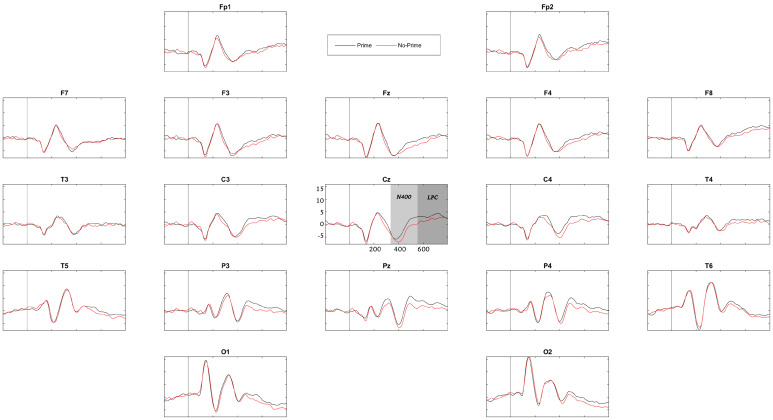
ERPs of the LD group. The N400 time range (355–550 ms) is represented in the Cz channel in pale gray. The LPC time range (555–800 ms) is represented in dark gray.

**Figure 5 brainsci-13-01022-f005:**
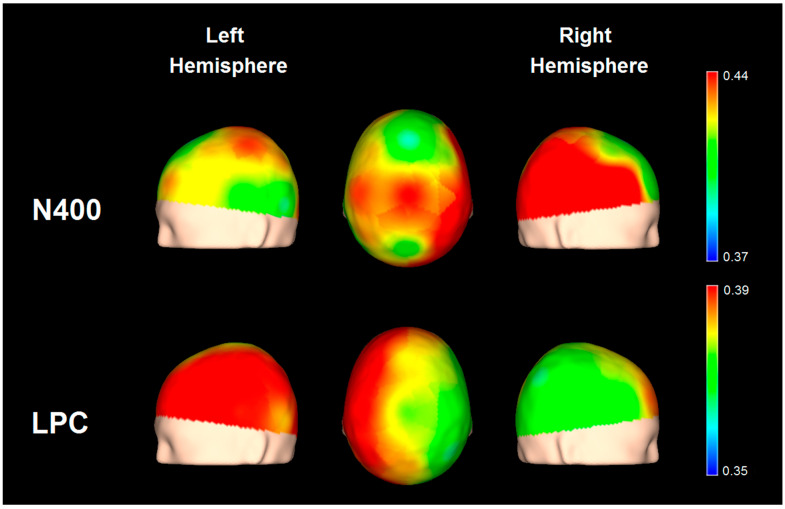
Statistical maps based on the rho values for the Verbal Comprehension Index as predictor of the N400 and LPC effects in the LD group.

**Table 1 brainsci-13-01022-t001:** Between-group permutation-based t-test comparison of the main descriptive characteristics of the participants.

	Control Group *n* = 12 Mean (SD)	LD Group *n* = 28 Mean (SD)	*t*	*p*
Age	9.08 (0.79)	9.11 (0.79)	−0.09	NS
WISC-IV: Full-Scale IQ	108.92 (10.57)	91.64 (11.24)	4.65	<0.001
Reading	67.31 (11)	30.98 (20.21)	5.99	<0.001
Writing	72.22 (11.16)	42 (17.48)	5.65	<0.001
Mathematics	71.67 (6.85)	36.11 (21.30)	5.77	<0.001

Note: NS = not significant; the variables of reading, writing, and mathematics are composite percentile scores from the Infant Neuropsychological Scale for Children.

**Table 2 brainsci-13-01022-t002:** Between-group permutation-based t-test comparison of the WISC-IV indices.

	Control Group *n* = 12 Mean (SD)	LD Group *n* = 28 Mean (SD)	*t*	*p*
Verbal Comprehension	115.75 (16.62)	90.39 (10.86)	5.89	<0.001
Perceptual Reasoning	106.92 (13.78)	97.43 (14.13)	2.01	NS
Working Memory	103.50 (8.67)	93.14 (12.20)	2.72	0.026
Processing Speed	95.25 (12.40)	92.21 (12.32)	0.73	NS

Note: NS = not significant.

**Table 3 brainsci-13-01022-t003:** Within-condition permutation-based t-test comparison of the behavioral results of the lexical decision task for each group.

Variable	Group	Mean Primed (SD)	Mean Non-Primed (SD)	*t*	*p*
Correct Responses (%)	Control	94.04 (4.87)	92.39 (5.14)	−1.97	0.037
	LD	88.92 (7.23)	87.48 (7.23)	−1.12	NS
Response Times (ms)	Control	757.17 (131.29)	826.58 (172.50)	3.33	0.001
	LD	814.36 (162.15)	865.96 (161.53)	3.79	<0.001

Note: NS = not significant.

**Table 4 brainsci-13-01022-t004:** Permutation-based regression analysis of the WISC-IV indices as independent regressors with the N400 and LPC components for all 19 channels in the LD group.

Subcomponent	Statistics	VC	PR	WM	PS
N400 Effect	Range of r^2^ values	[0.154, 0.221]	[0.089, 0.118]	[0.014, 0.038]	[0.005, 0.016]
	Range of slope values	[0.101, 0.128]	[0.062, 0.069]	[0.028, 0.047]	[0.015, 0.031]
	*p*	<0.027	NS	NS	NS
LPC Effect	Range of r^2^ values	[0.133, 0.161]	[0.063, 0.094]	[0.017, 0.038]	[0.073, 0.096]
	Range of slope values	[0.087, 0.098]	[0.046, 0.057]	[0.028, 0.042]	[0.057, 0.067]
	*p*	<0.042	NS	NS	NS

Note: NS = not significant; VC = Verbal Comprehension Index; PR = Perceptual Reasoning Index; WM = Working Memory Index; PS = Processing Speed Index.

## Data Availability

The dataset analyzed for this study will be uploaded to a cloud-based communal repository. The accession numbers will be provided during the review process.

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
