# Peer review of "Semantic Priming and Its Link to Verbal Comprehension and Working Memory in Children with Learning Disorders"

_brainsci, 2023, doi:10.3390/brainsci13071022_

Round 1

Reviewer 1 Report

the paper SEMANTIC PRIMING, VERBAL COMPREHENSION AND 

WORKING MEMORY ABILITY IN CHILDREN WITH  LEARNING DISORDERS is interesting, even though it was probably written very quickly. There are many mistakes both in formal English and when describing more scientific terms, especially electrophysiological ones. For example, see in the abstract 'an N400', the term 'school-age children', and the phrase 'The implications of these findings are discussed.' And so in all the text, I suggest reviewing all the English forms. Furthermore, the authors use the term 'electrical' for EEG, please, review all the text in 'electrophysiological'. Moreover please, explain better, in the text,  the differences between the N400 effect and N400 components (citing the right and more updated literature).

The authors write 'academic impairment'. What do the authors mean? I think the word must be reviewed in the text and in the image (Fig.1) . I think the authors could use the appropriate terminology. 

The ERP statistical analysis doesn't consider ROI; usually, according to the literature, is not advisable to carry out the analyzes on all EEGv channels, because this creates a bias (see Luck SJ, Gaspelin N. How to get statistically significant effects in any ERP experiment (and why you shouldn't). Psychophysiology. 2017 Jan;54(1):146-157). Moreover, the authors don't show the difference between groups. Please, review the statistical analysis in a way that shows also the 'between' differences. 

he paper SEMANTIC PRIMING, VERBAL COMPREHENSION AND 

WORKING MEMORY ABILITY IN CHILDREN WITH  LEARNING DISORDERS is interesting, even though it was probably written very quickly. There are many mistakes both in formal English and when describing more scientific terms, especially electrophysiological ones. For example, see in the abstract 'an N400', the term 'school-age children', and the phrase 'The implications of these findings are discussed.' And so in all the text, I suggest reviewing all the English forms. Furthermore, the authors use the term 'electrical' for EEG, please, review all the text in 'electrophysiological'. I suggest a professional review of English

Author Response

GENERAL CHANGES:

  • As prompted by reviewer 1, several terms have been corrected, together with extensive English editing (as also suggested by both reviewers). A main term has been introduced (Late Positive Complex-LPC) to replace the subpar term of N400b.
  • The title of the paper has been changed, as suggested by Reviewer 2.
  • The paper has been updated with 8 new and appropriate references.
  • A table and two figures have been added as response to Reviewer 2.
  • As indicated by the Editor, a Conclusion section has been added.

REVIEWER 1: “The paper is interesting, even though it was probably written very quickly.”

Q1. There are many mistakes both in formal English and when describing more scientific terms, especially electrophysiological ones. For example, see in the abstract 'an N400', the term 'school-age children', and the phrase 'The implications of these findings are discussed.' And so in all the text, I suggest reviewing all the English forms. The authors use the term 'electrical' for EEG, please, review all the text in 'electrophysiological’. The authors write 'academic impairment'. What do the authors mean? The word must be reviewed in the text and in the image (Fig.1) . I think the authors could use the appropriate terminology.

A: The above-mentioned terms have been corrected, and the abstract has been partially rewritten.

Q2. Explain better, in the text,  the differences between the N400 effect and N400 components (citing the right and more updated literature).

A: Both terms have been clarified in the Introduction. (See Lines: 79-85; 91-107.)

Q3. The ERP statistical analysis doesn't consider ROI; usually, according to the literature, is not advisable to carry out the analyzes on all EEG channels, because this creates a bias (see Luck & Gaspelin, 2017).

A: In the Introduction we have updated some facts regarding ROIs and the semantic priming literature of both children with LD and adults. Also, in the section 2.4 of the Method we added the following sentence: “The topography of the priming-related components in children with LD has previously shown notable variability compared to age-matched controls [33, 44], not to mention compared to the priming literature in healthy adults. Thus, we did not select any ROIs in order to capture the variance across all channels simultaneously analyzed.” (See lines: 110-117; 247-250.)

Q4. The authors don't show the difference between groups. Please, review the statistical analysis in a way that shows also the 'between' differences.

A: We apologize for this omission. A between-group analysis has been conducted and showed no significant differences. A new paragraph in Results now reads: “The between-group comparison did not reveal significant differences in the ERP components. This might be due to considerable variability in the brain responses of children with subtypes of LD [44, 61]; thus, some possible significant differences may have been canceled. Additionally, the control group consisted of only 12 children and this particular between-group comparison might have been underpowered. The within-condition results point to a picture in which the control group shows an expected semantic priming effect that is deficient in the LD group. This, coupled with the fact that the LD group did show an impaired performance in the indices of Verbal Comprehension and Working Memory, does not affect the aim of the following regression analysis of the LD group.” (Lines: 308-317.)

Reviewer 2 Report

This study investigated the relationship between working memory and the N400 effect in children with learning disorders. The relationships between the WISC-IV cognitive indices and the ERPs in prime and no-prime conditions were studied using a lexical decision task. It’s interesting that the findings have differences from those of previous studies. The authors found that verbal comprehension rather than working memory held a crucial role in semantic priming effect in children with LDs. The results are valuable for the study on the neural underpinnings of LDs and rehabilitation of children with LDs. However, there are still some problems to be clarified or improved before publication as follows:

Q1. The results could be more detailed. For example, to explore the brain regions where negative N400 components are more significant, and ERP-averaged topographies across the time window of 350-900 ms.

Q2. As for the discussion, the reasons why working memory, verbal comprehension, perceptual reasoning, and processing speed have significant or not significant effects could be specified. More comparisions or contrasts with previous studies concerning N400 and LPC are expected.

Q3. Concerning the experiment design, the scores of subject’s WISC-IV cognitive indices and the experimental flow chart should be added.

Q4 The current title of the manuscript needs revision as semantic priming seems not parallel with the other two, verbal comprehension and working memory which are two indices of WISC-IV Test.

Q5. Many expressions need to be improved grammatically, such as non-subject clauses, and the usage of semicolons and conjunctions.

Author Response

GENERAL CHANGES:

  • As prompted by reviewer 1, several terms have been corrected, together with extensive English editing (as also suggested by both reviewers). A main term has been introduced (Late Positive Complex-LPC) to replace the subpar term of N400b.
  • The title of the paper has been changed, as suggested by Reviewer 2.
  • The paper has been updated with 8 new and appropriate references.
  • A table and two figures have been added as response to Reviewer 2.
  • As indicated by the Editor, a Conclusion section has been added.

REVIEWER 2: “This study investigated the relationship between working memory and the N400 effect in children with learning disorders. The relationships between the WISC-IV cognitive indices and the ERPs in prime and no-prime conditions were studied using a lexical decision task. It’s interesting that the findings have differences from those of previous studies. The authors found that verbal comprehension rather than working memory held a crucial role in semantic priming effect in children with LDs. The results are valuable for the study on the neural underpinnings of LDs and rehabilitation of children with LDs. However, there are still some problems to be clarified or improved before publication as follows:”

Q1. The results could be more detailed. For example, to explore the brain regions where negative N400 components are more significant, and ERP-averaged topographies across the time window of 350-900 ms.

A: We apologize for the notable omission of information regarding topography, this was partly due to the nature of the findings: since in LD children the semantic priming effects are broader and more diffuse compared to healthy samples of subjects. We have added such facts in the introduction, method, and results, and with the inclusion of a new figure 5. (See lines: 91-98; 110-117; 247-250; 328-335.)

Q2. As for the discussion, the reasons why working memory, verbal comprehension, perceptual reasoning, and processing speed have significant or not significant effects could be specified. More comparisons or contrasts with previous studies concerning N400 and LPC are expected.

A: Now we have made more explicit the fact that the semantic priming effect has been previously linked to both Verbal Comprehension and Working Memory (and we thus failed to find a relationship with Working Memory in the current study). The other two WISC-IV indices of perceptual reasoning and processing speed are justified as a type of control with no expectation of significant relationship with the ERPs, and indeed they weren´t impaired in the LD group compared to controls nor were predictors of the ERP components. (See Lines: 125-133; 193-197; 378-409.)

Q3. Concerning the experiment design, the scores of subject’s WISC-IV cognitive indices and the experimental flow chart should be added.

A: This has been done as requested, with the new Figure 2 in Method, and Table 2 in Results.

Q4. The current title of the manuscript needs revision as semantic priming seems not parallel with the other two, verbal comprehension and working memory which are two indices of WISC-IV Test.

A: Thank you for this highly relevant suggestion, the title has indeed been modified.

Q5. Many expressions need to be improved grammatically, such as non-subject clauses, and the usage of semicolons and conjunctions.

A: The manuscript has been extensively corrected with the help of an English editing service.

Round 2

Reviewer 1 Report

Please, change the term Academic. The children can't have academic impairments. Please, use the right terms. 

Moreover, please, change the ERP Analysis using ROI. 

English must be necessary reviewed

Reviewer 2 Report

Compared to the previous version, this one contains a great deal of progress. There is a clearer explanation of how the experiments were conducted and the results are more comprehensive, particularly the analysis of LPC. Furthermore, the addition of Figure 5 allows the reader to visualize the degree of activation across different regions of the brain more intuitively. Nevertheless, there will be a minor revision concerning the following issues before publication:

Q1. In conjunction with the literature reviewed in the introduction, the discussion must be explored in depth. Why might the results of this study differ from those of previous studies?

Q2. Considering the experimental design, why is it necessary to add pseudo words that can induce N400 as well?

A minor revision on English language is also needed. 
